# Differentially Private Analysis for Binary Response Models: Optimality, Estimation, and Inference

Ce Zhang [1]   Yixin Han [1]   Yafei Wang [1]   Xiaodong Yan [2]   Linglong Kong [1]   Ting Li [3]   Bei Jiang [1]

## Abstract

Randomized response (RR) mechanisms constitute a fundamental and effective technique for ensuring label differential privacy (LabelDP). However, existing RR methods primarily focus on the response labels while overlooking the influence of covariates and often do not fully address optimality. To address these challenges, this paper explores optimal LabelDP procedures using RR mechanisms, focusing on achieving optimal estimation and inference in binary response models. We first analyze the asymptotic behaviors of RR binary response models and then optimize the procedure by maximizing the trace of the Fisher Information Matrix within the $\varepsilon$- and $(\varepsilon, \delta)$-LabelDP constraints. Our theoretical results indicate that the proposed methods achieve optimal LabelDP guarantees while maintaining statistical accuracy in binary response models under mild conditions. Furthermore, we develop private confidence intervals with nominal coverage for statistical inference. Extensive simulation studies and real-world applications confirm that our methods outperform existing approaches in terms of precise estimation, privacy protection, and reliable inference.

## 1. Introduction

**Motivation.** Privacy protection is crucial in modern statistical analysis and machine learning to handle personal information and sensitive data. Differential privacy (DP) (Dwork et al., 2006; 2014) has recently emerged as the gold standard for preserving privacy when analyzing sensitive data. It provides a statistical mechanism that enables users to analyze data without directly accessing sensitive information while exploring theoretical guarantees to prevent the exposure of private details. On the other hand, the random response mechanism (RR) (Warner, 1965) has gained significant attention in sensitive binary data and achieves effective DP guarantees by flipping the binary labels with a predetermined probability (Hout et al., 2010; Coutts & Jann, 2011; Kirchner, 2015; Oberski & Kreuter, 2020). Although a surge of works explores DP guarantees under RR mechanisms (Holohan et al., 2017; Fox et al., 2018; Ghazi et al., 2020; Pastore & Gastpar, 2021; Xu et al., 2023), there is still limited guidance on the optimal DP mechanism for RR labels, particularly in the context of estimation and inference for binary response models. To address this gap, we introduce a novel optimal DP RR mechanism tailored for binary response models that take into account the influence of covariates. Our proposed DP RR mechanism operates under the setting that the input covariates are publicly available while the response labels are highly sensitive, which requires rigorous protection. Motivated by the optimal design of the experiment via maximizing the various types of "data information" (Pukelsheim, 2006; Dette et al., 2012; Yang et al., 2013; Waite & Woods, 2022; Duarte et al., 2022), we primarily focus on exploring the optimal estimation and inference for the RR binary response models by maximizing the trace of the Fisher Information Matrix within a LabelDP feasible region for RR parameters. This ensures the optimality of the RR mechanism in safeguarding privacy while maintaining statistical accuracy.

**Literature review.** Numerous approaches have been proposed to develop DP algorithms through RR mechanisms (Ding et al., 2020; Garcelon et al., 2021; Biswas et al., 2023). In this section, we organize the relevant work into three groups and highlight how our methodology differs significantly, emphasizing the importance of a thorough review.

*DP via RR.* Based on the classical RR mechanisms, Kairouz et al. (2016) studied the optimal local DP mechanisms for binary data. Holohan et al. (2017) extended Warner's original technique (Warner, 1965) to establish an RR mechanism that simultaneously achieves DP and optimality by minimizing the variance estimator of the proportion of true answers.

*Equal contribution [1]Department of Mathematical and Statistical Sciences, University of Alberta, Edmonton, Canada [2]School of Mathematics and Statistics, Xi'an Jiaotong University, Xi'an, China [3]Department of Applied Mathematics, The Hong Kong Polytechnic University, Hong Kong, China. Correspondence to: Bei Jiang <bei1@ualberta.ca>, Ting Li <tingeric.li@polyu.edu.hk>.

*Proceedings of the 42nd International Conference on Machine Learning*, Vancouver, Canada. PMLR 267, 2025. Copyright 2025 by the author(s).

Furthermore, Ghazi et al. (2021) introduced a deep learning method to construct an RR mechanism. Recently, Waudby-Smith et al. (2023) presented a nonparametric extension of Warner's seminal RR mechanism tailored for bounded data with sequential interactivity. Furthermore, Barnes et al. (2020) and Asoodeh & Zhang (2022) investigated how fisher information from statistical samples can scale with the privacy parameter $\varepsilon$ under local DP constraints. Despite the success of DP via RR in various domains, the challenges of statistical inference for RR binary response models under DP remain unresolved, potentially impacting both accuracy and efficiency.

*Label differential privacy.* Label differential privacy (LabelDP) is a well-established method to protect sensitive labels while allowing non-private covariates under the RR mechanism in classification tasks (Chaudhuri & Hsu, 2011). LabelDP has received significant attention due to the increasing demands in various fields (Nayak & Adeshiyan, 2009; Wang et al., 2016; Busa-Fekete et al., 2021; Chaudhuri & Hsu, 2011; Busa-Fekete et al., 2021; Xu et al., 2023). More recently, Busa-Fekete et al. (2023) established the LabelDP mechanisms to share training data in machine learning, enabling accurate predictive model learning while safeguarding the privacy of each user's labels.

*Inference under RR binary response models.* In statistics, inference involves the use of representative training data to learn about population characteristics, often represented by model parameters (Tutz, 2011). Statistical inference under LabelDP goes beyond providing point estimates; its goal is to construct *confidence intervals* that capture the true values of the model/population parameters while accounting for sampling variability and noise introduced by LabelDP (Ghazi et al., 2022). Numerous works have been performed on the estimation of binary response models using RR mechanisms without inference (Van den Hout et al., 2007; Hsieh et al., 2010; Duan et al., 2016). Although Fox et al. (2018) has explored statistical inference and extended the estimate to a wide range of RR designs, their method does not consider optimal estimation and inference within the LabelDP framework, which is the main objective of our paper.

**Our goals and contributions.** Traditional optimal LabelDP RR mechanisms, such as those in Holohan et al. (2017) and Wang et al. (2017), focus solely on optimizing privacy for the response variable $Y$, but ignore the role of covariates $X$. In contrast, our approach incorporates $X$ by maximizing the trace of the Fisher Information Matrix in the context of RR binary response models, leading to improved statistical efficiency and accuracy – capabilities that traditional methods do not offer. Specifically, we make the following major contributions.

- This is the first work to investigate the optimal $\varepsilon$- and $(\varepsilon, \delta)$-LabelDP mechanisms using RR binary response models. Specifically, our definition of optimality aims to maximize the trace of the asymptotic information matrix in (5) within the feasible LabelDP region for the RR parameters, as illustrated in Figure 1. Furthermore, we establish the optimal inference for RR binary response models with a specified level of confidence within the LabelDP framework.

- We provide a rigorous theoretical analysis of the proposed method, including the privacy guarantees and asymptotic normality. Theoretical results establish that this novel optimal LabelDP RR mechanism guarantees $\varepsilon$- and $(\varepsilon, \delta)$-LabelDP privacy while achieving optimal estimators and asymptotic normality for the regression coefficients in RR binary response models. Additionally, we develop a privatized confidence interval that delivers asymptotically valid coverage while preserving privacy.

- We conduct extensive simulations and real-data experiments to demonstrate that our approach outperforms several existing mechanisms, providing significantly higher efficiency regarding estimation accuracy and coverage probability. Notably, our method achieves a threefold improvement in coverage probability within the confidence interval, closely matching the performance of non-private methods. This highlights the effectiveness of our approach in balancing privacy preservation with high statistical efficiency.

**Paper organization.** The rest of this paper is organized as follows. Section 2 reviews the essential background of LabelDP and RR mechanisms. We introduce the RR binary response models and establish the novel optimal LabelDP RR mechanism under the $\varepsilon$- and $(\varepsilon, \delta)$-LabelDP frameworks in Section 3. Section 4 provides the theoretical results on LabelDP guarantees and the private inference of the confidence interval. Extensive simulation studies are conducted in Section 5 to demonstrate the superior performance of the proposed methods. In Section 6, we apply our proposed method to a real-world dataset focused on measuring plagiarism among students to further illustrate its effectiveness. Section 7 concludes the paper with several additional topics. The detailed technical proofs are included in the appendix.

## 2. Preliminaries

In this section, we present a brief review of the LabelDP and RR mechanisms, which lay the groundwork for our proposed method.

**LabelDP.** DP has emerged as a state-of-the-art framework for releasing privacy-preserving data in data science and machine learning (Dwork et al., 2006; 2014; Dong et al., 2022). The core of DP is to ensure that an individual's data

is sufficiently masked within the dataset. This enables analysis to be performed without direct access to the original data, offering theoretical assurances that prevent the exposure of sensitive information. Our work focuses mainly on the binary response model, so we introduce the concept of LabelDP, a specialized application of DP that specifically targets the response space.

**Definition 2.1** $((\varepsilon, \delta)$-LabelDP (Ghazi et al., 2021)). Let $\mathcal{X}$ and $\mathcal{Y}$ denote the covariate and binary response spaces, respectively. A randomized algorithm or mechanism $\mathcal{A} : \mathcal{X} \to \mathcal{Y}$ is $(\varepsilon, \delta)$-LabelDP for some parameters $\varepsilon > 0$, $0 \le \delta \le 1$. If for any pair of datasets $\mathcal{D}, \mathcal{D}'$ that only differ in the label of a single example and any measurable event $\mathcal{C} \subseteq \mathcal{Y}$, it holds that

$$\mathbb{P}\left(\mathcal{A}(\mathcal{D}) \in \mathcal{C}\right) \le e^{\varepsilon} \mathbb{P}\left(\mathcal{A}(\mathcal{D}') \in \mathcal{C}\right) + \delta,$$

where the probability $\mathbb{P}$ is taken over the randomness of $\mathcal{A}$.

The privacy budget $(\varepsilon, \delta)$ quantifies the level of protection of privacy and the probability of relaxation. If $\delta = 0$, $\mathcal{A}$ is called pure LabelDP, denoted by $\varepsilon$-LabelDP; if $\delta > 0$, it is considered the approximate LabelDP. Generally speaking, the smaller the sizes of both $\varepsilon$ and $\delta$, the stronger the protection level, but this will limit the utility of the estimators. Commonly used LabelDP mechanisms randomize target quantities (e.g., parameter, loss, gradient, or aggregate statistics) by adding random noises derived from elaborately designed distributions to hide each individual's contribution (Kairouz et al., 2016).

**RR mechanism.** Suppose that there are $n$ individuals, each with an observed binary response $Y_i \in \{0, 1\}$, dependent on the true value $Y_i^* \in \{0, 1\}$. The definition 2.2 describes the relationship between $Y$ and $Y^*$ for the RR mechanism via a design matrix.

**Definition 2.2** (Design matrix). The RR mechanism for a binary true value $Y_i^*$ follows a $2 \times 2$ design matrix

$$P = \left( \begin{array}{cc} p_{00} & p_{01} \\ p_{10} & p_{11} \end{array} \right) = \left( \begin{array}{cc} p_{00} & 1 - p_{00} \\ 1 - p_{11} & p_{11} \end{array} \right),$$

where $p_{kj} = \mathbb{P}(Y_i = j \mid Y_i^* = k)$ with $j, k = \{0, 1\}$, represents the conditional response probability (Wang et al., 2016; Holohan et al., 2017; Chaudhuri & Mukerjee, 2020), and adheres the fact that $p_{00} + p_{01} = 1, p_{10} + p_{11} = 1$.

Subsequently, the probability mass function of each observed value $Y_i$ can be calculated by

$$\begin{aligned} \mathbb{P}\left(Y_i = 0\right) &= \{1 - \mathbb{P}\left(Y_i^* = 1\right)\} p_{00} + \mathbb{P}\left(Y_i^* = 1\right)\left(1 - p_{11}\right) \\ &= p_{00} - \mathbb{P}\left(Y_i^* = 1\right)\left(p_{00} + p_{11} - 1\right), \\ \mathbb{P}\left(Y_i = 1\right) &= \mathbb{P}\left(Y_i^* = 1\right) p_{11} + \{1 - \mathbb{P}\left(Y_i^* = 1\right)\}\left(1 - p_{00}\right) \\ &= 1 - p_{00} + \left(p_{00} + p_{11} - 1\right) \mathbb{P}\left(Y_i^* = 1\right). \end{aligned} \quad (1)$$

In scenarios where both $p_{00}$ and $p_{11}$ are equal to 1, the RR mechanism directly reflects the true response, resulting in $\mathbb{P}(Y_i = 0) = \mathbb{P}(Y_i^* = 0)$ and $\mathbb{P}(Y_i = 1) = \mathbb{P}(Y_i^* = 1)$. However, this RR mechanism only addresses the unsupervised aspect of the response. This means that it does not take into account the relationship between the response variable and the covariates or features, leading to potential inefficiencies and inaccuracies when used in supervised learning contexts, such as regression models. In the following sections, we will investigate the application of RR binary response models within the framework of LabelDP. This approach provides a deeper insight into the effective application of privacy-preserving techniques in binary response models.

## 3. Methodology

In this section, we aim to develop a novel LabelDP RR mechanism that yields an optimal estimator for the RR binary response models while safeguarding the privacy of individual responses. We first introduce a new data structure that incorporates RR and binary response models. Then we present the fundamental framework of the proposed optimal LabelDP RR mechanism.

### 3.1. RR binary response models

Suppose we observe $n$ independent and identical distributed (i.i.d.) observations $\mathcal{D}_n^* = \{\boldsymbol{X}_i, Y_i^*\}_{i=1}^n$ from a binary response model

$$\mathbb{E}\left(Y_i^* \mid \boldsymbol{X}_i\right) = \mathbb{P}\left(Y_i^* = 1 \mid \boldsymbol{X}_i\right) = G\left(\boldsymbol{\beta}_*^\top \boldsymbol{X}_i\right), \quad (2)$$

where $\boldsymbol{X}_i = (X_{i1}, \ldots, X_{id})^\top \in \mathbb{R}^d$ is the $d$-dimensional covariate, $Y_i^* \in \{0, 1\}$ is the true binary label, $\boldsymbol{\beta}_*$ is the true parameter vector. Throughout this paper, we assume that only the responses $Y_i^*$ contain sensitive information, while the covariates $\boldsymbol{X}_i$ are considered non-sensitive. The function $G(\boldsymbol{\beta}_*^\top \boldsymbol{X}_i)$ denotes the link function, and the inverse of the link function is typically assumed to be monotonic and differentiable to the linear predictor $\boldsymbol{\beta}_*^\top \boldsymbol{X}_i$. Depending on the form of $G(\boldsymbol{\beta}_*^\top \boldsymbol{X}_i)$, the Model (2) can accommodate various binary response models as special cases, including Logistic regression $\exp(\boldsymbol{\beta}_*^\top \boldsymbol{X}_i)/\{1 + \exp(\boldsymbol{\beta}_*^\top \boldsymbol{X}_i)\}$, Cauchy distribution $\pi^{-1} \arctan(\boldsymbol{\beta}_*^\top \boldsymbol{X}_i) + 1/2$, and Probit regression $\Phi(\boldsymbol{\beta}_*^\top \boldsymbol{X}_i)$, where $\Phi$ typically represents the cumulative distribution function of the standard normal distribution, as explained in (Tutz, 2011; McCullagh, 2019).

Due to privacy considerations, researchers often encounter a practical challenge: they typically only have access to a potentially privatized dataset $\mathcal{D}_n = \{\boldsymbol{X}_i, Y_i\}_{i=1}^n$ instead of the true dataset $\mathcal{D}_n^*$ when using the RR mechanism. In particular, the labels in $\mathcal{D}_n$ may not accurately represent the true responses. In such cases, it is crucial to utilize a calibration counterpart binary response to construct the RR mechanism, as exemplified in (1) and (2). Consequently, we

formulate the RR binary response models as

$$\mathbb{E}\left(Y_i \mid \boldsymbol{X}_i\right) = \mathbb{P}\left(Y_i = 1 \mid \boldsymbol{X}_i\right)$$
$$= 1 - p_{00} + (p_{00} + p_{11} - 1)\, G\left(\boldsymbol{\beta}_*^\top \boldsymbol{X}_i\right). \quad (3)$$

Note that Equation (1) provides a straightforward relationship between Model (2) and Model (3), effectively adapting (2) to account for the randomness introduced by the RR mechanism. As a result, there is no need for a debiasing step when estimating Model (3), since it already includes the necessary adjustments due to (1).

The Maximum likelihood estimator (MLE) is effective in estimating $\boldsymbol{\beta}_*$ because the maximum likelihood equation for Model (3) shares a structure similar to the maximum likelihood equations of the generalized linear model (GLM) maximum likelihood equations (Blair et al., 2015). To get the MLE of $\boldsymbol{\beta}_*$ based on the observed $\mathcal{D}_n$, we solve the following optimization problem

$$\widehat{\boldsymbol{\beta}} = \underset{\boldsymbol{\beta}_*}{\arg\max}\, l_n(\boldsymbol{\beta}_*)$$
$$= \underset{\boldsymbol{\beta}_*}{\arg\max}\, \frac{1}{n} \sum_{i=1}^{n} \left\{ Y_i \log \frac{p_i}{1 - p_i} + \log\left(1 - p_i\right) \right\}, \quad (4)$$

where $p_i = \mathbb{P}\left(Y_i = 1 \mid \boldsymbol{X}_i\right)$, $\eta_i = \boldsymbol{\beta}_*^\top \boldsymbol{X}_i$, and $l_n(\boldsymbol{\beta}_*)$ is the log-likelihood function. Let $S(\cdot)$ be the derivative of $l_n(\cdot)$ and $\mathbb{I}(\cdot)$ be the Fisher Information Matrix with respect to $\boldsymbol{\beta}$, respectively. The $S(\cdot)$ and $\mathbb{I}(\cdot)$ in $\mathcal{D}_n$ can be specified as follows:

$$S(\boldsymbol{\beta}; p_{00}, p_{11}) = \frac{1}{n} \sum_{i=1}^{n} \frac{(p_{00} + p_{11} - 1)(Y_i - p_i)}{p_i(1 - p_i)} \frac{\partial G(\eta_i)}{\partial \eta_i} \boldsymbol{X}_i,$$

$$\mathbb{I}(\boldsymbol{\beta}; p_{00}, p_{11}) = \frac{1}{n} \sum_{i=1}^{n} \frac{(p_{00} + p_{11} - 1)^2}{p_i(1 - p_i)} \left\{ \frac{\partial G(\eta_i)}{\partial \eta_i} \right\}^2 \boldsymbol{X}_i \boldsymbol{X}_i^\top. \quad (5)$$

The derivative $S$ and the Fisher Information Matrix $\mathbb{I}$ play crucial roles in determining the asymptotic behaviour of the estimator, which is essential for establishing the optimality of the DP RR mechanism.

### 3.2. Optimal LabelDP mechanism

We establish the optimal LabelDP mechanisms for the RR binary response models (3) in this subsection. First, we construct the feasible regions of RR parameters $(p_{00}, p_{11})$ under LabelDP constraints in Definition 3.1 inspired by (Holohan et al., 2017). Figure 1 illustrates the corresponding feasible regions.

**Definition 3.1.** An RR mechanism (1) satisfies $\varepsilon$-LabelDP if $(p_{00}, p_{11}) \in \mathcal{R} \subset \mathbb{R}^2$, or satisfies $(\varepsilon, \delta)$-LabelDP if $(p_{00}, p_{11}) \in \mathcal{R}' \subset \mathbb{R}^2$, where $\mathcal{R}$ and $\mathcal{R}'$ are the LabelDP

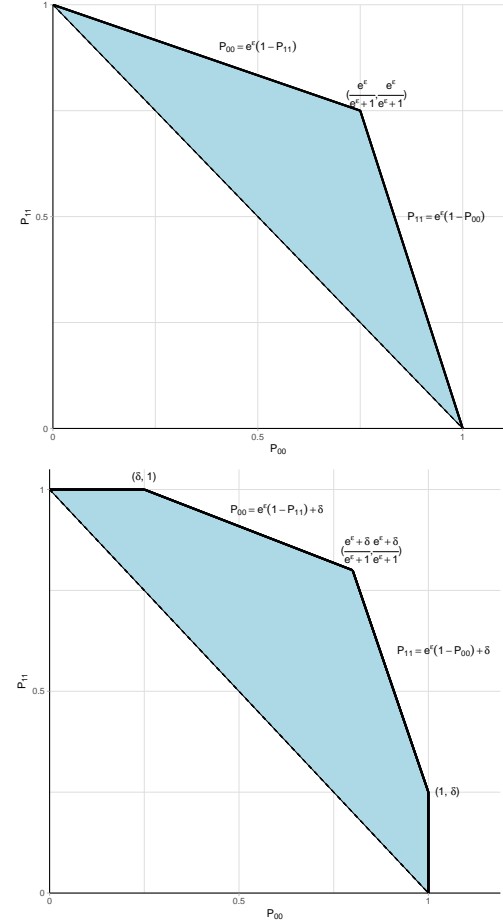

Figure 1. Feasible regions of the RR mechanisms. Top panel: $\varepsilon$-LabelDP; Bottom panel: $(\varepsilon, \delta)$-LabelDP.

feasible regions for RR parameters $(p_{00}, p_{11})$, defined as:

$$\mathcal{R} = \begin{cases} p_{00}, p_{11} \leq 1, \\ p_{00} + p_{11} > 1, \\ p_{00} \leq e^{\varepsilon}(1 - p_{11}), \\ p_{11} \leq e^{\varepsilon}(1 - p_{00}) \end{cases}, \quad \mathcal{R}' = \begin{cases} p_{00}, p_{11} \leq 1, \\ p_{00} + p_{11} > 1, \\ p_{00} \leq e^{\varepsilon}(1 - p_{11}) + \delta, \\ p_{11} \leq e^{\varepsilon}(1 - p_{00}) + \delta \end{cases}.$$

In general, a default assumption is that the RR mechanism tends to favor the true value, that is, $p_{00}, p_{11} > 0.5$ (Wang, 2015). Setting $p_{00}, p_{11} > 0.5$ implies that the RR mechanism is more likely to generate a true answer. This assumption is crucial for the statistical analysis of Model (3), as the RR mechanism's tendency to lean towards the truth enables more meaningful insights.

Next, we develop our optimal LabelDP mechanisms for the RR binary response model. To achieve our goal, we borrow the concepts from the design of experiments and consider the $T$-optimality criterion (Pukelsheim, 2006; Dette et al., 2012), which maximizes the trace of the Fisher Information Matrix $\mathcal{M}(\boldsymbol{\beta}; p_{00}, p_{11}) = \mathrm{tr}[\mathbb{I}(\boldsymbol{\beta}; p_{00}, p_{11})]$, restricted to

the LabelDP feasible regions $\mathcal{R}$ and $\mathcal{R}'$,

$$(\widehat{p}_{00}, \widehat{p}_{11}) = \underset{(p_{00}, p_{11}) \in \mathcal{R} \text{ or } \mathcal{R}'}{\arg\max} \mathcal{M}(\boldsymbol{\beta}; p_{00}, p_{11}). \quad (6)$$

It is important to note that while both $T$-optimality and $D$-optimality aim at maximizing information, $D$-optimality involves maximizing the determinant of the Fisher Information Matrix, which is computationally challenging and more complex due to the involvement of determinant calculations, especially for high-dimensional parameter spaces. However, $T$-optimality is computationally simpler, making it significantly easier to implement in practical scenarios without sacrificing the quality of inference. Moreover, in many cases, the two criteria are equivalent or closely related, and maximizing the trace often provides a satisfactory approximation to maximizing the determinant.

*Remark* 3.2. In detail, our $T$-optimality criterion is equivalent in asymptotic efficiency but far more practical. The trace decomposes into a sum of variances (diagonal elements), reducing the problem to scalar optimization (Lemma 4.2). This allows closed-form solutions for $(p_{00}, p_{11})$ on the boundary of $\mathcal{R}$ (Theorems 4.1 and 4.7). The trace directly corresponds to minimizing the average variance of $\widehat{\boldsymbol{\beta}}$, which aligns with our goal of precise estimation. Both criteria yield consistent estimators, but $T$-optimality achieves this with $O(d)$ complexity versus $O(d^3)$ for $D$-optimality due to determinant calculations, where $d$ is the number of covariates.

*Remark* 3.3. Awan & Slavkovic (2019) optimized the RR mechanisms solely for the binary response $Y$ by minimizing $\mathrm{Var}(\widehat{\sum Y})$, where $\sum Y$ is the sample sum, which is a complete sufficient statistic for the binomial model in their paper, without considering covariate information $X$. while our Fisher information maximization $\mathbb{I}(\boldsymbol{\beta}; p_{00}, p_{11})$ explicitly incorporates covariate $X$ through the model (3). Our novel method explicitly leverages the covariate structure, enhancing the estimation efficiency when covariates are present.

The estimators $(\widehat{p}_{00}, \widehat{p}_{11})$ do not depend on $\boldsymbol{\beta}$ based on the specific formulation of $\mathcal{M}(\boldsymbol{\beta}; p_{00}, p_{11})$. This independence arises because $\mathcal{M}(\boldsymbol{\beta}; p_{00}, p_{11})$ is designed to isolate $p_{00}$ and $p_{11}$ from the influence of $\boldsymbol{\beta}$ on $\partial G(\eta_i)/\partial \eta_i$. Therefore, in practical applications, the initial step is to use $\mathcal{M}(\boldsymbol{\beta}; p_{00}, p_{11})$ to obtain the optimal estimators of $p_{00}$ and $p_{11}$. Once these values are estimated, we then plug them into (4) to obtain $\widehat{\boldsymbol{\beta}}$. The estimated $\widehat{\boldsymbol{\beta}}$ is optimal in the sense that it maximizes the trace of the Fisher Information Matrix $\mathbb{I}(\boldsymbol{\beta}; p_{00}, p_{11})$.

# 4. Theoretical results

In this section, we start by establishing the theoretical guarantees for two LabelDP mechanisms, followed by the construction of the privatized confidence interval for our proposed methods. We begin with the theoretical guarantees for $\varepsilon$-LabelDP and then extend our focus to $(\varepsilon, \delta)$-LabelDP. Furthermore, we explore the statistical inference of $\widehat{\boldsymbol{\beta}}$ based on Model (3). To support our theoretical results, we introduce several technical assumptions necessary for the proofs.

**Assumption 4.1.** $\mathcal{M}(\boldsymbol{\beta}; p_{00}, p_{11})$ is twice continuously differentiable with respect to $p_{00}$ and $p_{11}$.

**Assumption 4.2.** Suppose that $p_{00} + p_{11} > 1$.

Assumption 4.1 is the regular conditions imposed to ensure the unique and finite maximum in optimization (Boyd & Vandenberghe, 2004). Assumption 4.2 requires that the RR parameters satisfy $p_{00} + p_{11} > 1$ to ensure the identifiability of the parameter for the RR binary response model (3) across the feasible regions.

*Remark* 4.3. Assumption 4.2 serves multiple purposes by improving theoretical robustness and optimality. First, it ensures that the RR mechanism tends to capture the true response accurately, thereby improving the utility of the privatized response label data. Second, it restricts the feasible region for $(p_{00}, p_{11})$, leading to a well-defined optimization process and practical solutions for optimal LabelDP RR mechanisms.

**Lemma 4.4.** *Suppose Assumptions 4.1–4.2 hold. There exists $(p_{00}^*, p_{11}^*)$ such that $(p_{00}^*, p_{11}^*) \in \arg\max_{(p_{00}, p_{11}) \in \mathcal{R} \text{ or } \mathcal{R}'} \mathcal{M}(\boldsymbol{\beta}; p_{00}, p_{11})$, where $(p_{00}^*, p_{11}^*)$ lie on the boundaries of $\mathcal{R}$ or $\mathcal{R}'$.*

Lemma 4.4 demonstrates that $\mathcal{M}(\boldsymbol{\beta}; p_{00}, p_{11})$ is convex and thus confirms the existence of optimal solutions for the RR parameters in binary response models. The optimal $(p_{00}^*, p_{11}^*)$ attains on the boundaries of the feasible regions $\mathcal{R}$ and $\mathcal{R}'$ by maximizing the function $\mathcal{M}(\boldsymbol{\beta}; p_{00}, p_{11})$. Under Lemma 4.4, we will proceed to establish the optimal RR mechanisms for $\varepsilon$- and $(\varepsilon, \delta)$-LabelDP constraints.

### 4.1. Optimal mechanism for $\varepsilon$-LabelDP

**Theorem 4.5** ($\varepsilon$-LabelDP)**.** *Suppose Assumptions 4.1–4.2 hold. Given $\varepsilon > 0$, the $\varepsilon$-LabelDP RR mechanism which maximizes $\mathcal{M}(\boldsymbol{\beta}; p_{00}, p_{11})$ is given by the privacy design matrix*

$$P_\varepsilon = \begin{pmatrix} \frac{e^\varepsilon}{e^\varepsilon + 1} & \frac{1}{e^\varepsilon + 1} \\ \frac{1}{e^\varepsilon + 1} & \frac{e^\varepsilon}{e^\varepsilon + 1} \end{pmatrix}.$$

It is crucial to point out that the traditional optimal LabelDP mechanisms, such as those discussed in (Holohan et al., 2017; Wang et al., 2017), primarily focus on optimizing privacy based solely on the response variable $Y$. These approaches typically aim to minimize the variance of the estimator related to $Y$, without incorporating the additional information provided by covariates. In contrast, while our results are similar to Theorem 2 from (Holohan et al., 2017)

when establishing an optimal mechanism for $\varepsilon$-LabelDP, our method starts by optimizing the RR binary response model itself, specifically by maximizing the trace of the Fisher Information Matrix. This optimization inherently incorporates covariates ($X$), which are essential for enhancing the statistical efficiency and accuracy of the estimates in the binary response model, which is a significant advantage that traditional LabelDP methods do not provide.

### 4.2. Optimal mechanism for $(\varepsilon, \delta)$-LabelDP

The $(\varepsilon, \delta)$-LabelDP is a relaxation of $\varepsilon$-LabelDP when $\delta > 0$. In this section, we utilize the following parameterization to explore the optimal $(\varepsilon, \delta)$-LabelDP RR mechanism. We first parametrize $p_{11} = e^\varepsilon(1-p_{00}) + \delta$ in $\mathcal{R}'$ as $p_{00} = r_\delta(t)$ and $p_{11} = s_\delta(t)$, where

$$r_\delta(t) = \left(1 + e^{-\varepsilon}\delta\right)(1-t) + \frac{e^\varepsilon + \delta}{e^\varepsilon + 1}t, \text{ and } s_\delta(t) = \frac{e^\varepsilon + \delta}{e^\varepsilon + 1}t,$$

for $t \in (0, 1]$. By the symmetry of $p_{00}$ and $p_{11}$ based on the form of $p_{00} = e^\varepsilon(1 - p_{11}) + \delta$, we can then parameterize the line $p_{00} = e^\varepsilon(1 - p_{11}) + \delta$ in $\mathcal{R}'$ as $p_{00} = s_\delta(t)$ and $p_{11} = r_\delta(t)$. We start with the following lemma:

**Lemma 4.6.** *Suppose Assumptions 4.1–4.2 hold. For any $\delta > 0$ and constants $l \leq u$, with $l, u \in (0, 1]$, it follows that*

$$\arg \max_{t \in [l,u]} \mathcal{M}(\boldsymbol{\beta}; r_\delta(t), s_\delta(t)) \subseteq \{l, u\}.$$

Lemma 4.6 presents that the optimal solution of $(p_{00}, p_{11})$ exists under $(\varepsilon, \delta)$-LabelDP constraint, which enables us to establish the corresponding optimal LabelDP RR mechanism in Theorem 4.7.

**Theorem 4.7** ($(\varepsilon, \delta)$-LabelDP). *Suppose Assumptions 4.1–4.2 hold. Given $\varepsilon > 0$ and $\delta \in (0, 1)$, the $(\varepsilon, \delta)$-LabelDP RR mechanisms which maximizes $\mathcal{M}(\boldsymbol{\beta}; p_{00}, p_{11})$ are described as follows:*

(i) *If $\mathcal{M}(\boldsymbol{\beta}; \frac{e^\varepsilon + \delta}{e^\varepsilon + 1}, \frac{e^\varepsilon + \delta}{e^\varepsilon + 1}) \geq \mathcal{M}(\boldsymbol{\beta}; 1, \delta)$ and $\mathcal{M}(\boldsymbol{\beta}; \frac{e^\varepsilon + \delta}{e^\varepsilon + 1}, \frac{e^\varepsilon + \delta}{e^\varepsilon + 1}) \geq \mathcal{M}(\boldsymbol{\beta}; \delta, 1)$, then the privacy design matrix is $P_{\varepsilon, \delta} = \begin{pmatrix} \frac{e^\varepsilon + \delta}{e^\varepsilon + 1} & \frac{1 - \delta}{e^\varepsilon + 1} \\ \frac{1 - \delta}{e^\varepsilon + 1} & \frac{e^\varepsilon + \delta}{e^\varepsilon + 1} \end{pmatrix}$.*

(ii) *If $\mathcal{M}(\boldsymbol{\beta}; 1, \delta) \geq \mathcal{M}(\boldsymbol{\beta}; \frac{e^\varepsilon + \delta}{e^\varepsilon + 1}, \frac{e^\varepsilon + \delta}{e^\varepsilon + 1})$ and $\mathcal{M}(\boldsymbol{\beta}; 1, \delta) \geq \mathcal{M}(\boldsymbol{\beta}; \delta, 1)$, then the privacy design matrix is $P_{\varepsilon, \delta} = \begin{pmatrix} 1 & 0 \\ 1 - \delta & \delta \end{pmatrix}$.*

(iii) *If $\mathcal{M}(\boldsymbol{\beta}; \delta, 1) \geq \mathcal{M}(\boldsymbol{\beta}; \frac{e^\varepsilon + \delta}{e^\varepsilon + 1}, \frac{e^\varepsilon + \delta}{e^\varepsilon + 1})$ and $\mathcal{M}(\boldsymbol{\beta}; \delta, 1) \geq \mathcal{M}(\boldsymbol{\beta}; 1, \delta)$, then the privacy design matrix is $P_{\varepsilon, \delta} = \begin{pmatrix} \delta & 1 - \delta \\ 0 & 1 \end{pmatrix}$.*

In Theorem 4.7, the optimal $(\varepsilon, \delta)$-LabelDP design matrix is determined through three steps in a real-world application. First, we plug in the sets of values $(p_{00}, p_{11}) = (1, \delta), (\delta, 1)$, and $(\frac{e^\varepsilon + \delta}{e^\varepsilon + 1}, \frac{e^\varepsilon + \delta}{e^\varepsilon + 1})$ into formula (4). These inputs allow us to estimate the corresponding values of the parameter $\boldsymbol{\beta}_*$ for each case. Once $\widehat{\boldsymbol{\beta}}$ is estimated, we then calculate the value of $\mathcal{M}(\widehat{\boldsymbol{\beta}}; p_{00}, p_{11})$ for each of these sets of probability pairs. The final step involves comparing these calculated values of $\mathcal{M}(\widehat{\boldsymbol{\beta}}; p_{00}, p_{11})$ to determine which set of $p_{00}$ and $p_{11}$ provides the optimal outcome.

It is important to highlight that the above procedure does not require direct access to private data for selecting matrices. Instead, it relies on theoretical estimates and comparisons based on predetermined probability pairs $(p_{00}, p_{11})$ within the LabelDP framework, thereby preserving privacy throughout the selection process. Theorem 4.7 constructs the optimal estimator of the true $\boldsymbol{\beta}_*$ while safeguarding $(\varepsilon, \delta)$-LabelDP guarantee in binary response models in the same spirit as Theorem 4.5. Consequently, our mechanisms enable investigators to release sensitive response label data with higher statistical accuracy.

### 4.3. Inference with privacy

Private statistical inference is important as well as estimation. In this section, we provide the construction of the privacy-preserving confidence interval (CI) for each coefficient $\beta_{*j}$ under $\varepsilon$- and $(\varepsilon, \delta)$-LabelDP mechanisms. The following additional assumptions are imposed to facilitate the technical proof of asymptotic normality for MLE.

**Assumption 4.8.** The true parameter $\boldsymbol{\beta}_*$ must be identifiable. The log likelihood function $l_n(\boldsymbol{\beta}_*)$ is convex and can be differentiated twice continuously over $\boldsymbol{\beta}_* \in \Theta$.

**Assumption 4.9.** The information matrix $\mathbb{I}(\boldsymbol{\beta}_*)$ is positive definite at $\boldsymbol{\beta}_*$.

Assumptions 4.8 and 4.9 are standard in asymptotic inference. Assumption 4.8 (on convexity and smoothness of the log-likelihood) holds in most binary regression settings, and Assumption 4.9 (positive definiteness of the Fisher Information Matrix) ensures model identifiability, a common condition in GLMs. While asymptotic results assume large samples, these assumptions often hold well in practice, even with moderate sample sizes–as supported by our real data analysis.

**Theorem 4.10** (Asymptotic Normality). *Suppose Assumptions 4.2 and 4.8–4.9 hold. As $n \to \infty$, $\widehat{\boldsymbol{\beta}}$ is asymptotically normally distributed as*

$$\sqrt{n}(\widehat{\boldsymbol{\beta}} - \boldsymbol{\beta}_*) N\left(0, \mathbb{I}(\boldsymbol{\beta}_*)^{-1}\right).$$

The unbiasedness and asymptotic normality in Theorem

4.10 are crucial for constructing confidence intervals on each cared coefficient $\beta_{*j}, 1 \leq j \leq d$ in Model (3) under the LabelDP constraints.

Beyond point estimation, our framework enables formal statistical inference with privacy guarantees. Specifically, Corollary 4.11 establishes the asymptotic valid confidence intervals of each coefficient $\beta_{*j}$ under the $\varepsilon$- and $(\varepsilon, \delta)$-LabelDP mechanisms, containing $\beta_{*j}$ with high probability (e.g., 0.95) as sample size $n \to \infty$.

**Corollary 4.11.** *Suppose the Assumptions in Theorem 4.10 hold. Denote by $\alpha$ the target significant level. For $1 \leq j \leq d$, the DP $(1 - \alpha)$-confidence interval for $\beta_{*j}$ is*

$$\mathrm{CI}_{1-\alpha}(\beta_{*j}) = \left[\widehat{\beta}_j - \mathbb{I}^{-1/2}(\widehat{\beta}_j)z_{1-\frac{\alpha}{2}}, \widehat{\beta}_j + \mathbb{I}^{-1/2}(\widehat{\beta}_j)z_{1-\frac{\alpha}{2}}\right],$$

*where $z_{1-\alpha}$ is the $(1 - \alpha)$-the quantile of the standard normal distribution. Moreover, as $n \to \infty$, the nominal coverage probability of the confidence interval $\mathrm{CI}_{1-\alpha}(\beta_{*j})$ is*

$$\lim_{n \to \infty} \mathbb{P}\left(\widehat{\beta}_j \in \mathrm{CI}_{1-\alpha}(\beta_{*j})\right) = 1 - \alpha.$$

Corollary 4.11 establishes that our privatized confidence interval achieves asymptotically valid coverage while maintaining privacy protection. Although Theorem 4.10 and Corollary 4.11 are rooted in established likelihood theory, their application in our context is vital to validate our simulation results, particularly in terms of coverage probability. In particular, while our results are similar to Theorem 2 of (Holohan et al., 2017) in establishing an optimal mechanism for $\varepsilon$-LabelDP, our approach extends beyond by enabling private statistical inference for RR binary response models using both $\varepsilon$- and $(\varepsilon, \delta)$-LabelDP mechanisms. This represents a significant advantage over traditional optimal LabelDP methods (Holohan et al., 2017; Wang et al., 2017), which typically do not support such a direct and effective approach to private statistical inference.

# 5. Simulation studies

In this section, we conduct extensive numerical experiments to evaluate the finite sample performance of the proposed mechanism with several competing methods when fitting RR binary response.

## 5.1. Settings

We consider several benchmarks for comparison with our proposed methods.
**NP.** Non-DP RR binary response (Fox et al., 2018), which estimates $\beta_*$ without any privacy consideration.
**RRbR.** Apply the traditional DP RR mechanism (Holohan et al., 2017) solely to the response variable, then fit a binary

response model with this privatized response. The confidence intervals for RRbR are built by fitting the same binary response model as our method, but using responses privatized by the traditional RR mechanism. Standard asymptotic techniques are then applied to construct the intervals. However, this ignores the extra bias and variability introduced by randomization and fails to use covariate information to correct for these effects.
**ORRbR.** Optimal LabelDP RR binary response models (the proposed methods).
We set $d = 4$, sample size $n = 10^5$ and fix $\beta_* = (1, 0.25, 0, 0.5)^\top$. Two covariate structures are considered.
**Scenario I** *(Independence structure).* The covariate $X = (x_1, x_2, x_3, x_4)^\top$ is i.i.d. generated with the intercept $x_1 = 1$, $x_2 \sim N(0, 1)$, $x_3 \sim N(0, 1.5^2)$, and $x_4 \sim N(0, 0.5^2)$.
**Scenario II** *(Dependence structure).* The covariate $X = (x_1, x_2, x_3, x_4)^\top$ is i.i.d. generated with the intercept $x_1 = 1$, and $(x_2, x_3, x_4) \sim N(0, \Sigma)$ with an autoregressive dependence structure $(\Sigma)_{jl} = 0.5^{|j-l|}$, $2 \leq j, l \leq d$.
In each scenario, two link functions are considered, including Logistic regression $G(\beta_*^\top X_i) = \exp(\beta_*^\top X_i)/\{1 + \exp(\beta_*^\top X_i)\}$ and Probit regression $G(\beta_*^\top X_i) = \Phi(\beta_*^\top X_i)$. We set the nominal significant level $\alpha = 0.05$. We study both the $\varepsilon$-DP ($\delta = 0$) and $(\varepsilon, \delta)$-DP ($\delta = 10^{-5}$). All simulation results are based on $B = 500$ independent replications, and $\widehat{\beta}^{(b)}$ is the estimator obtained from the $b$-th replication. The performance of the proposed methods is evaluated along with the above benchmarks through comparisons of estimation (mean squared error, $\mathrm{MSE}(\beta_*) = B^{-1} \sum_{b=1}^{B} \|\widehat{\beta}^{(b)} - \beta_*\|^2$), privacy $(\varepsilon, \delta)$, and inference (coverage probability of $1$-$\alpha$ confidence interval).

## 5.2. Results

Figures 2–5 present the empirical MSE of $\widehat{\beta}$ and the coverage probability of an approximate 95% confidence interval for $\beta_*$ for Scenarios I–II against different privacy parameters under $\varepsilon$- and $(\varepsilon, \delta)$-LabelDP constraints, respectively. Specifically, the top subfigures in each of these figures correspond to $\delta = 0$, while the bottom ones correspond to $\delta = 10^{-5}$. In Figures 2 and 4, the $y$-axis shows the logarithm of Mean Squared Error (MSE), and in Figures 3 and 5, it shows the empirical average coverage probability (CP) for an approximate 95% confidence interval of the true parameter $\beta_*$, with the $x$-axis representing the privacy parameter $\varepsilon$ in all cases. The chosen privacy budget values $\varepsilon$ are $\{0.05, 0.06, 0.07, 0.08, 0.09, 0.1, 0.5, 0.7, 1\}$, which are not evenly spaced or set according to a log scale. Instead, these values were deliberately selected in an irregular and somewhat random manner to extensively test the robustness of our method across a wide range of privacy constraints.

All figures demonstrate that NP possesses the smallest MSE and the most accurate coverage probability as expected, since no privacy is considered. Remarkably, our ORRbR

shows a higher coverage probability when $\varepsilon$ is exceedingly small due to the conservative nature of privacy-preserving estimators under stringent privacy constraints. Hence, under tight privacy constraints, the ORRbR method introduces significant noise to protect privacy. This could lead to overly conservative estimates, resulting in an increase in coverage probability compared to the NP method. As the privacy budget $\varepsilon$ increases, all comparing methods sacrifice accuracy for better privacy protection, consistent with theoretical analysis. Moreover, our proposed ORRbR outperforms the RRbR with a smaller MSE and a higher coverage probability due to the presence of covariate information in ORRbR.

In addition, the performance of our proposed approach has improved as the privacy parameter $\varepsilon$ has increased for both values of $\delta$. For example, in the top left plot of Figure 2, the log MSE is approximately 0 for $\varepsilon = 0.05$ and $\delta = 0$, while it is approximately -15 for $\varepsilon = 0.7$ and $\delta = 0$. Additionally, analogous results are depicted in the right panels of Figures 4 for Probit regression under Scenario II. Improvements were observed as $\varepsilon$ increased for both $\varepsilon$- and $(\varepsilon, \delta)$-LabelDP constraints. Furthermore, when comparing the panels in Figures 2 and 4, as well as Figures 3 and 5, we can observe significant differences between Scenarios I and II, particularly in terms of mean squared error (MSE) and coverage probability. Scenario II exhibits a larger MSE compared to Scenario I. Additionally, the coverage probability in Scenario II appears to be less accurate than in Scenario I. These disparities can be attributed to the underlying structural differences between the two scenarios.

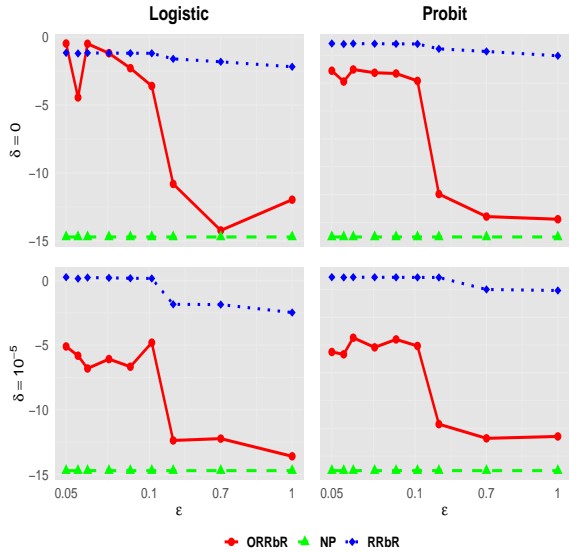

Figure 2. Empirical log MSE for three methods against different $\varepsilon$ under Scenario I.

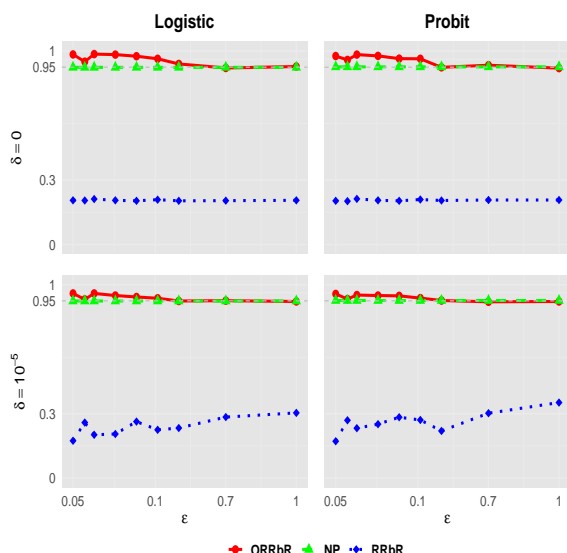

Figure 3. Empirical average coverage probability (CP) for three methods against different $\varepsilon$ under Scenario I.

## 6. Real-data application

In this section, we demonstrate the practical utility of our proposed method by applying it to a real-world dataset focused on measuring plagiarism among students, as detailed in (Jann et al., 2012). This dataset comprises responses from 474 German and Swiss students who were surveyed using either direct questioning or the crosswise (CW) technique to estimate the prevalence of plagiarism. For our analysis, the dataset was divided into two parts: 75% of the data was used to train the model, while the remaining 25% was reserved for testing. The covariates in our model include three key variables: *pp* (indicating whether the question pertains to partial or severe plagiarism), *RR* (denoting whether the crosswise method or direct questioning was used), and *age* (the student's age). These variables are essential to capture the nuances of the data and were included in the model to assess their impact on responses.

We implemented three methods – NP, RRbR, and ORRbR, as used in our simulation studies – to predict the probabilities of different responses. The performance of these models was assessed by calculating the MSE between the predicted probabilities and the actual responses in the test set. The results presented in Figure 6 highlight the trade-off between differential privacy and statistical accuracy. Compared to RRbR, the lower MSE values achieved by the proposed ORRbR method at various $\varepsilon$ levels demonstrate its superior performance. In particular, as $\varepsilon$ increases, the MSE for OR-RbR decreases even further, highlighting its effectiveness in balancing privacy and accuracy.

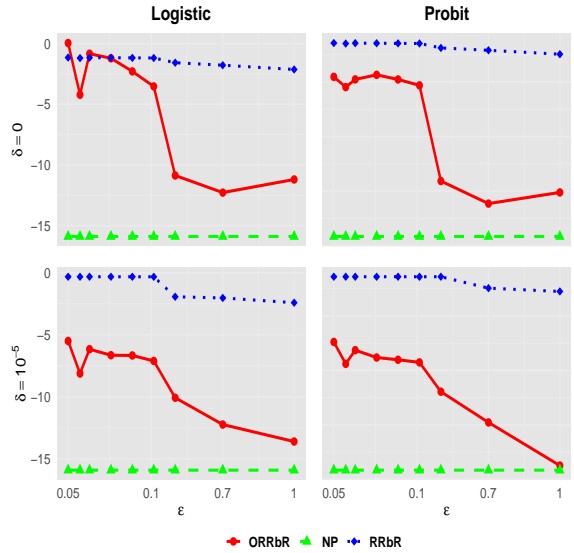
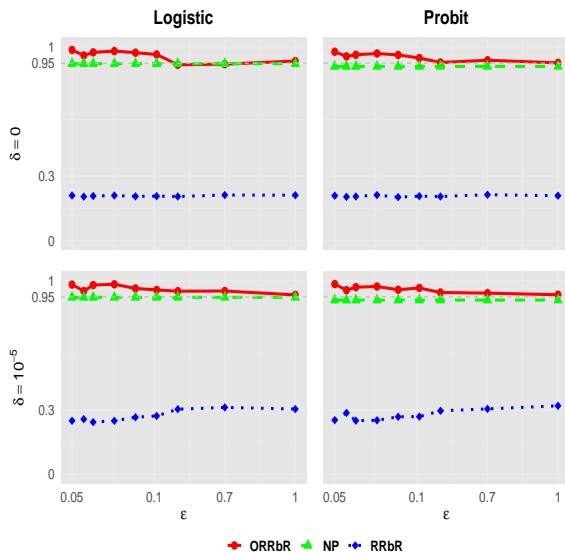

*Figure 4.* Empirical log MSE for three methods against different $\varepsilon$ under Scenario II.

*Figure 5.* Empirical average coverage probability (CP) for three methods against different $\varepsilon$ under Scenario II.

## 7. Conclusion

This paper proposed two RR mechanisms with an optimal design for binary response models under $\varepsilon$- and $(\varepsilon, \delta)$-LabelDP constraints. Theoretical and numerical results reveal that the proposed optimal LabelDP RR mechanism successfully outperforms the traditional optimal DP RR mechanism (Holohan et al., 2017) in terms of estimation, privacy, and statistical inference when applied to fit a binary response model with privatized responses. Starting with parametric models like GLMs is methodologically natural, as they offer well-defined Fisher Information and their analytical tractability allows for rigorous theoretical development. This mirrors the trajectory of many pioneering works in DP, which initially focused on basic statistical tasks such as mean and median estimation, before extending to more structured models like linear and logistic regression (e.g., Kulkarni et al. (2021); Narayanan et al. (2022); Asi et al. (2022); Alparslan et al. (2023); Kulesza et al. (2023); Brown et al. (2024)). This balance between privacy protection and statistical efficiency marks a significant advancement in private data analysis. Our approach allows researchers to confidently release sensitive binary response data while improving statistical efficiency with strong privacy guarantees.

We conclude this paper by acknowledging several limitations and suggesting directions for future research. First, our method can be naturally extended to multiclass outcomes using the $k$-RR mechanism. The optimization strategy of maximizing the trace of the Fisher Information Matrix remains valid for multiclass scenarios, as discussed in Yao & Wang (2019). Specifically, for $k$-class responses, the de-

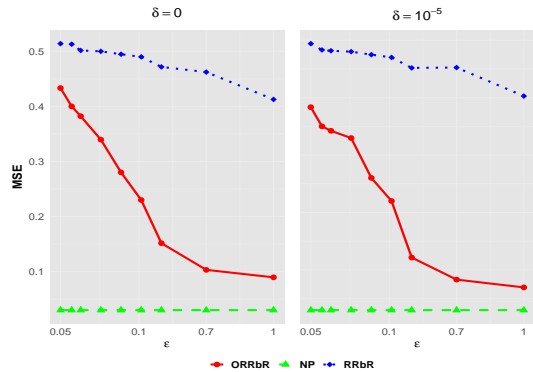

*Figure 6.* Empirical MSE for three methods against different $\varepsilon$ under the real-data application.

sign matrix in Definition 2.2 of our paper becomes a $k \times k$ stochastic matrix, and the optimality criterion would involve maximizing the Fisher Information's trace across multiple response probabilities under LabelDP constraints. This is an exciting direction that we are actively exploring, and preliminary theoretical analysis indicates promising scalability. We plan to provide more detailed theoretical developments and numerical evaluations in future work. Second, extending our method to other concepts of privacy, such as Rényi DP (Mironov, 2017) or Gaussian DP (Dong et al., 2022), remains challenging due to the unclear design of the feasible region in our equation (3). Lastly, investigating other optimal criteria, such as minimizing label classification errors, deserves further exploration to enhance the robustness and applicability of our approach.

## Acknowledgements

We would like to thank the anonymous reviewers and area chairs for great feedback on the paper. Xiaodong Yan was supported by the National Key R&D Program of China (No. 2023YFA1008701) and the National Natural Science Foundation of China (No. 12371292) equally. Bei Jiang and Linglong Kong were partially supported by grants from the Canada CIFAR AI Chairs program, the Alberta Machine Intelligence Institute (AMII), and Natural Sciences and Engineering Council of Canada (NSERC), and Linglong Kong was also partially supported by grants from the Canada Research Chair program from NSERC.

## Impact statement

The broader impact of this work extends across multiple domains where privacy-preserving statistical inference is essential, including healthcare, social sciences, and finance. By providing optimal privacy protection while maintaining statistical accuracy, our method enables more reliable inference in sensitive applications such as medical diagnosis, online surveys, and secure machine learning. In scenarios where protecting individual responses is critical, our approach ensures that privacy guarantees do not come at the cost of statistical efficiency, allowing researchers and practitioners to draw meaningful conclusions from privatized data.

Additionally, our framework facilitates the construction of private confidence intervals, ensuring valid statistical inference under strict privacy constraints. This capability is particularly valuable for organizations and policymakers who require both privacy preservation and rigorous statistical validity in their analyses. Extensive empirical evaluations demonstrate that our method outperforms existing approaches in estimation precision, coverage probability, and computational efficiency. By striking a balance between privacy and statistical accuracy, our approach lays a strong foundation for advancing differentially private statistical modelling in real-world applications.

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

# Appendix

**Proof of Lemma 4.3.** Firstly, after some calculation on $\frac{\partial \mathcal{M}(\boldsymbol{\beta}; p_{00}, p_{11})}{\partial p_{00}}$ and $\frac{\partial^2 \mathcal{M}(\boldsymbol{\beta}_0; p_{00}, p_{11})}{\partial p_{00}^2}$, we have

$$
\begin{aligned}
\frac{\partial \mathcal{M}(\boldsymbol{\beta}; p_{00}, p_{11})}{\partial p_{00}} =& \frac{1}{n} \sum_{i=1}^{n} \sum_{j=1}^{d} \frac{(p_{00} + p_{11} - 1)\left[2p_i(1 - p_i) + (p_{00} + p_{11} - 1)(1 - 2p_i)(1 - G(\eta_i))\right]}{\left[p_i(1 - p_i)\right]^2} \\
& \cdot \left\{ \frac{\partial G(\eta_i)}{\partial \eta_i} \right\}^2 X_{ij}^2.
\end{aligned}
$$

$$
\begin{aligned}
\frac{\partial^2 \mathcal{M}(\boldsymbol{\beta}; p_{00}, p_{11})}{\partial p_{00}^2} =& \frac{2}{n} \sum_{i=1}^{n} \sum_{j=1}^{d} \left\{ \left[ p_i(1 - p_i) + (p_{00} + p_{11} - 1)(1 - 2p_i)(1 - G(\eta_i)) \right]^2 \right. \\
& \left. + (p_{00} + p_{11} - 1)^2 p_i(1 - p_i)(1 - G(\eta_i))^2 \right\} \frac{1}{\left[p_i(1 - p_i)\right]^3} \\
& \cdot \left\{ \frac{\partial G(\eta_i)}{\partial \eta_i} \right\}^2 X_{ij}^2.
\end{aligned}
$$

Note that $\frac{\partial^2 \mathcal{M}(\boldsymbol{\beta}; p_{00}, p_{11})}{\partial p_{00}^2} \geq 0$, it is easy to verify that $\frac{\partial \mathcal{M}(\boldsymbol{\beta}; p_{00}, p_{11})}{\partial p_{00}}$ increases as $p_{00}$ increases within the region $\mathcal{R}$ or $\mathcal{R}'$. We know that $p_{00}$ is greater than $1 - p_{11}$ under Assumption 4.2. Since $\frac{\partial \mathcal{M}(\boldsymbol{\beta}; 1 - p_{11}, p_{11})}{\partial p_{00}} = 0$, we can deduce that $\frac{\partial \mathcal{M}(\boldsymbol{\beta}; p_{00}, p_{11})}{\partial p_{00}} > 0$ when $p_{00} > 1 - p_{11}$. It follows that $\mathcal{M}(\boldsymbol{\beta}; p_{00}, p_{11})$ increases as $p_{00}$ increases.

Secondly, after performing some calculations, we have

$$
\begin{aligned}
\frac{\partial \mathcal{M}(\boldsymbol{\beta}; p_{00}, p_{11})}{\partial p_{11}} =& \frac{1}{n} \sum_{i=1}^{n} \sum_{j=1}^{d} \frac{(p_{00} + p_{11} - 1)\left[2p_i(1 - p_i) - (p_{00} + p_{11} - 1)(1 - 2p_i)G(\eta_i)\right]}{\left[p_i(1 - p_i)\right]^2} \\
& \cdot \left\{ \frac{\partial G(\eta_i)}{\partial \eta_i} \right\}^2 X_{ij}^2,
\end{aligned}
$$

$$
\begin{aligned}
\frac{\partial^2 \mathcal{M}(\boldsymbol{\beta}; p_{00}, p_{11})}{\partial p_{11}^2} =& \frac{2}{n} \sum_{i=1}^{n} \sum_{j=1}^{d} \left\{ \left[ p_i(1 - p_i) - (p_{00} + p_{11} - 1)(1 - 2p_i)G(\eta_i) \right]^2 \right. \\
& \left. + (p_{00} + p_{11} - 1)^2 p_i(1 - p_i) G_i^2 \right\} \frac{1}{\left[p_i(1 - p_i)\right]^3} \left\{ \frac{\partial G(\eta_i)}{\partial \eta_i} \right\}^2 X_{ij}^2.
\end{aligned}
$$

Similar to the reasoning above, we verify that $\mathcal{M}(\boldsymbol{\beta}; p_{00}, p_{11})$ is also an increasing function as $p_{11}$ increases since $\frac{\partial \mathcal{M}(\boldsymbol{\beta}; p_{00}, p_{11})}{\partial p_{11}} > 0$ when $p_{11} > 1 - p_{00}$. Thus, the maximum of $\mathcal{M}(\boldsymbol{\beta}; p_{00}, p_{11})$ for $(p_{00}, p_{11})$ is attained on the boundaries of the feasible regions $\mathcal{R}$ or $\mathcal{R}'$. $\qquad\square$

**Proof of Theorem 4.4.** Lemma 4.3 shows that the optimal mechanism occurs when the parameters $(p_{00}, p_{11})$ are on the boundaries of $\mathcal{R}$. Accordingly, at least one of the inequalities $p_{11} \leq e^{\varepsilon}(1 - p_{00})$ and $p_{00} \leq e^{\varepsilon}(1 - p_{11})$ must be tight (i.e., either $p_{11} = e^{\varepsilon}(1 - p_{00})$ or $p_{00} = e^{\varepsilon}(1 - p_{11})$), which can be listed as the following cases.

**Case 1.** Suppose $p_{11} \leq e^{\varepsilon}(1 - p_{00})$ holds. It implies that $p_{11} = e^{\varepsilon}(1 - p_{00})$ with the constraint that $p_{11}$ is non-negative and $p_{00} \leq e^{\varepsilon}(1 - p_{11})$. For $t \in (0, 1]$, we parameterize the line $p_{11} = e^{\varepsilon}(1 - p_{00})$ as $p_{00} = r(t)$ and $p_{11} = s(t)$ with the condition $p_{00} + p_{11} > 1$

$$
\begin{aligned}
r(t) &= (1 - t) + \frac{e^{\varepsilon}}{1 + e^{\varepsilon}} t = 1 - e^{-\varepsilon} s(t), \\
s(t) &= \frac{e^{\varepsilon}}{1 + e^{\varepsilon}} t.
\end{aligned}
\tag{A.1}
$$

See the proof of Theorem 2 in (Holohan et al., 2017) for details of the parameterization. After substituting $p_{00} = r(t)$ and

$p_{11} = s(t)$ into $\mathcal{M}(\boldsymbol{\beta}; p_{00}, p_{11})$, we have

$$\mathcal{M}\left(\boldsymbol{\beta}; r(t), s(t)\right) = \frac{1}{n} \sum_{i=1}^{n} \sum_{j=1}^{d} \frac{t(e^{\varepsilon} - 1)^2}{\left\{e^{\varepsilon} + 1 - t\left[1 + (e^{\varepsilon} - 1)G(\eta_i)\right]\right\}\left[1 + (e^{\varepsilon} - 1)G(\eta_i)\right]}$$
$$\cdot \left\{\frac{\partial G(\eta_i)}{\partial \eta_i}\right\}^2 X_{ij}^2,$$

$$\frac{\partial \mathcal{M}\left(\boldsymbol{\beta}; r(t), s(t)\right)}{\partial t} = \frac{1}{n} \sum_{i=1}^{n} \sum_{j=1}^{d} \frac{(e^{\varepsilon} - 1)^2 (e^{\varepsilon} + 1)\left[1 + (e^{\varepsilon} - 1)G(\eta_i)\right]}{\left\{e^{\varepsilon} + 1 - t\left[1 + (e^{\varepsilon} - 1)G(\eta_i)\right]\right\}^2 \left[1 + (e^{\varepsilon} - 1)G(\eta_i)\right]^2}$$
$$\cdot \left\{\frac{\partial G(\eta_i)}{\partial \eta_i}\right\}^2 X_{ij}^2.$$

Given $e^{\varepsilon} > 1$, we have $\frac{\partial \mathcal{M}(\boldsymbol{\beta}; r(t), s(t))}{\partial t} > 0$, which implies that the maximum value of $t$ maximizing $\mathcal{M}(\boldsymbol{\beta}; r(t), s(t))$ lies in the interval $(0, 1]$ and is equal to 1, i.e., $p_{00}^{\varepsilon} = p_{11}^{\varepsilon} = 1/(e^{\varepsilon} + 1)$. Therefore, we conclude that the privacy design matrix for the $\varepsilon$-LabelDP RR mechanism while maximizing $\mathcal{M}(\boldsymbol{\beta}; p_{00}, p_{11})$ is

$$P_{\varepsilon} = \begin{pmatrix} p_{00}^{\varepsilon} & 1 - p_{00}^{\varepsilon} \\ 1 - p_{11}^{\varepsilon} & p_{11}^{\varepsilon} \end{pmatrix} = \begin{pmatrix} \frac{e^{\varepsilon}}{e^{\varepsilon}+1} & \frac{1}{e^{\varepsilon}+1} \\ \frac{1}{e^{\varepsilon}+1} & \frac{e^{\varepsilon}}{e^{\varepsilon}+1} \end{pmatrix}.$$

**Case 2.** Let $p_{00} \leq e^{\varepsilon}(1 - p_{11})$ tight that means $p_{00} = e^{\varepsilon}(1 - p_{11})$, constrained by $p_{00} \geq 0$ and $p_{11} \leq e^{\varepsilon}(1 - p_{00})$. By symmetry of the equations $p_{00} \leq e^{\varepsilon}(1 - p_{11})$ and $p_{11} \leq e^{\varepsilon}(1 - p_{00})$, we define $p_{00} = s(t)$ and $p_{11} = r(t)$. By the similar proof of Case 1, we obtain the same privacy design matrix as shown in Case 1. $\qquad \square$

**Proof of Lemma 4.5**. As the Proof of Theorem 4.4, we parameterize the line $p_{11} = e^{\epsilon}(1 - p_{00}) + \delta$ of the region $\mathcal{R}'$ as $p_{00} = r_{\delta}(t)$ and $p_{11} = s_{\delta}(t)$ :

$$r_{\delta}(t) = \left(1 + e^{-\epsilon}\delta\right)(1 - t) + \frac{e^{\epsilon} + \delta}{e^{\epsilon} + 1}t, \tag{A.2}$$

$$s_{\delta}(t) = \frac{e^{\epsilon} + \delta}{e^{\epsilon} + 1}t, \tag{A.3}$$

for $t \in (0, 1]$. By symmetry, we can parameterize the line $p_{00} = e^{\epsilon}(1 - p_{11}) + \delta$ of the region $\mathcal{R}'$ as $p_{00} = s_{\delta}(t)$ and $p_{11} = r_{\delta}(t)$. Hence, in the following, we only need to prove the result for the case that $p_{00} = r_{\delta}(t)$ and $p_{11} = s_{\delta}(t)$. By plugging (A.2) and (A.3) into $\mathcal{M}(\boldsymbol{\beta}; p_{00}, p_{11})$, we have

$$\mathcal{M}\left(\boldsymbol{\beta}; r_{\delta}(t), s_{\delta}(t)\right) = \frac{1}{n} \sum_{i=1}^{n} \sum_{j=1}^{d} \frac{\left[t(e^{\varepsilon} - 1)(1 + e^{-\varepsilon}\delta) + \delta + e^{-\varepsilon}\delta\right]^2}{\left\{t(1 + e^{-\varepsilon}\delta)\left[1 + (e^{\varepsilon} - 1)G(\eta_i)\right] - (\delta + e^{-\varepsilon}\delta)(1 - G(\eta_i))\right\}}$$
$$\cdot \frac{1}{\left\{(e^{\varepsilon} + 1)\left[1 + e^{-\varepsilon}\delta(1 - G(\eta_i))\right] - t(1 + e^{-\varepsilon}\delta)\left[1 + (e^{\varepsilon} - 1)G(\eta_i)\right]\right\}}$$
$$\cdot \left\{\frac{\partial G(\eta_i)}{\partial \eta_i}\right\}^2 X_{ij}^2.$$

Denote

$$A(t) = t(e^{\varepsilon} - 1)\left(1 + e^{-\varepsilon}\delta\right) + \delta + e^{-\varepsilon}\delta,$$
$$B_i(t) = t\left(1 + e^{-\varepsilon}\delta\right)\left[1 + (e^{\varepsilon} - 1)G(\eta_i)\right] - \left(\delta + e^{-\varepsilon}\delta\right)(1 - G(\eta_i)),$$
$$C_i(t) = (e^{\varepsilon} + 1)\left[1 + e^{-\varepsilon}\delta(1 - G(\eta_i))\right] - t\left(1 + e^{-\varepsilon}\delta\right)\left[1 + (e^{\varepsilon} - 1)G(\eta_i)\right].$$

Hence, we rewrite $\mathcal{M}\left(\boldsymbol{\beta};r_\delta(t),s_\delta(t)\right)=\frac{1}{n}\sum_{i=1}^{n}\sum_{j=1}^{d}\frac{A^2(t)}{B_i(t)C_i(t)}\left\{\frac{\partial G(\eta_i)}{\partial \eta_i}\right\}^2 X_{ij}^2$. And thus we have

$$
\begin{aligned}
\frac{\partial \mathcal{M}\left(\boldsymbol{\beta};r_\delta(t),s_\delta(t)\right)}{\partial t}=&\frac{1}{n}\sum_{i=1}^{n}\sum_{j=1}^{d}\left\{\frac{2A(t)B_i(t)C_i(t)\left(e^\varepsilon-1\right)\left(1+e^{-\varepsilon}\delta\right)}{B_i^2(t)C_i^2(t)}\right.\\
&\left.-\frac{A^2(t)\left[e^\varepsilon+1-2B_i(t)\right]\left(1+e^{-\varepsilon}\delta\right)\left[1+\left(e^\varepsilon-1\right)G\left(\eta_i\right)\right]}{B_i^2(t)C_i^2(t)}\right\}\\
&\cdot\left\{\frac{\partial G\left(\eta_i\right)}{\partial \eta_i}\right\}^2 X_{ij}^2,
\end{aligned}
$$

$$
\begin{aligned}
\frac{\partial^2 \mathcal{M}\left(\boldsymbol{\beta};r_\delta(t),s_\delta(t)\right)}{\partial t^2}=&\frac{1}{n}\sum_{i=1}^{n}\sum_{j=1}^{d}\left\{\frac{2B_i(t)C_i(t)\left(1+e^{-\varepsilon}\delta\right)}{B_i^4(t)C_i^4(t)}\right.\\
&\cdot\frac{\left\{\left(e^\varepsilon-1\right)B_i(t)C_i(t)-A(t)\left[e^\varepsilon+1-2B_i(t)\right]\left[1+\left(e^\varepsilon-1\right)G\left(\eta_i\right)\right]\right\}^2}{B_i^4(t)C_i^4(t)}\\
&\left.+\frac{2A^2(t)B_i^2(t)C_i^2(t)\left(1+e^{-\varepsilon}\delta\right)^2\left[1+\left(e^\varepsilon-1\right)G\left(\eta_i\right)\right]^2}{B_i^4(t)C_i^4(t)}\right\}\\
&\cdot\left\{\frac{\partial G\left(\eta_i\right)}{\partial \eta_i}\right\}^2 X_{ij}^2.
\end{aligned}
$$

Note that $\delta>0$ and $e^\varepsilon>1$, we have $\frac{\partial^2 \mathcal{M}(\boldsymbol{\beta};r_\delta(t),s_\delta(t))}{\partial t^2}\geq 0$. Then,

$$
\arg\max_{t\in[l,u]}\mathcal{M}\left(\boldsymbol{\beta};r_\delta(t),s_\delta(t)\right)\subseteq\{l,u\},
$$

which means that the maximal $\mathcal{M}\left(\boldsymbol{\beta};r_\delta(t),s_\delta(t)\right)$ on $t\in[l,u]$ will occur at one of its extreme points $l$ or $u$. $\qquad\square$

**Proof of Theorem 4.6.** By Lemma 4.5, we can see that an extreme point of $p_{00}=e^\varepsilon\left(1-p_{11}\right)+\delta$ is located at $t=1$, that $p_{00}^{\varepsilon,\delta}=p_{11}^{\varepsilon,\delta}=r_\delta(1)=s_\delta(1)=\frac{e^\varepsilon+\delta}{e^\varepsilon+1}$. Also, another extreme point can be found at $t_0(\varepsilon,\delta)=\frac{\delta(e^\varepsilon+1)}{e^\varepsilon+\delta}$, that, $p_{00}^{\varepsilon,\delta}=r_\delta(t_0)=1$ and $p_{11}^{\varepsilon,\delta}=s_\delta(t_0)=\delta$. Therefore, we obtain

$$
\arg\max_{t\in[t_0,1]}\mathcal{M}\left(\boldsymbol{\beta};r_\delta(t),s_\delta(t)\right)\subseteq\{t_0,1\}.
$$

Next, we aim to determine the sign of $\mathcal{M}\left(\boldsymbol{\beta};r_\delta(t_0),s_\delta(t_0)\right)-\mathcal{M}\left(\boldsymbol{\beta};r_\delta(1),s_\delta(1)\right)$. If $\mathcal{M}\left(\boldsymbol{\beta};r_\delta(t_0),s_\delta(t_0)\right)-\mathcal{M}\left(\boldsymbol{\beta};r_\delta(1),s_\delta(1)\right)>0$, then the optimal privacy design matrix is

$$
P_{\varepsilon,\delta}=\begin{pmatrix} 1 & 0 \\ 1-\delta & \delta \end{pmatrix},
$$

else the optimal privacy design matrix is

$$
P_{\varepsilon,\delta}=\begin{pmatrix} \frac{e^\varepsilon+\delta}{e^\varepsilon+1} & \frac{1-\delta}{e^\varepsilon+1} \\ \frac{1-\delta}{e^\varepsilon+1} & \frac{e^\varepsilon+\delta}{e^\varepsilon+1} \end{pmatrix}.
$$

Similarly, if we set $p_{11}=e^\varepsilon\left(1-p_{00}\right)+\delta$ to be tight, and parameterize $p_{11}=r_\delta(t)$ and $p_{00}=s_\delta(t)$. Then if $\mathcal{M}\left(\boldsymbol{\beta};s_\delta(t_0),r_\delta(t_0)\right)-\mathcal{M}\left(\boldsymbol{\beta};s_\delta(1),r_\delta(1)\right)>0$, then the optimal privacy design matrix is

$$
P_{\varepsilon,\delta}=\begin{pmatrix} \delta & 1-\delta \\ 0 & 1 \end{pmatrix},
$$

else the optimal privacy design matrix is

$$
P_{\varepsilon,\delta}=\begin{pmatrix} \frac{e^\varepsilon+\delta}{e^\varepsilon+1} & \frac{1-\delta}{e^\varepsilon+1} \\ \frac{1-\delta}{e^\varepsilon+1} & \frac{e^\varepsilon+\delta}{e^\varepsilon+1} \end{pmatrix},
$$

and accordingly, we claim the assertion. $\qquad\square$

