# OpenReview forum: "Differentially Private Analysis for Binary Response Models: Optimality, Estimation, and Inference"
_ICML.cc/2025/Conference — ICML 2025 poster_

### Official Review · Reviewer_rwTX · 2025-03-11

**Overall Recommendation:** 1

**Summary:**

This paper proposes a new method for ensuring label differential privacy in classification tasks through the randomized response mechanism with optimality guarantees. Furthermore,  the paper proposes differentially private confidence intervals based on the former method.

**Claims And Evidence:**

Most of the claims are supported by clear evidence. However, some concerns remain: (i) the Validity of Def. 3.1 (see below), and (ii) the meaning of "optimality". The paper states that the method borrows the T-optimality criterion. The criterion and the optimality guarantees, however, are never introduced. It remains unclear if the proposed method is indeed "optimal" (and the precise meaning thereof) or only an improvement over the standard RR mechanism.

**Essential References Not Discussed:**

/

**Experimental Designs Or Analyses:**

The scope of the experiments is limited and very simplistic. Furthermore, the results are not reproducible, as no code is provided for the experiments.

The evaluation of the empirical results is not coherent—for example, lines 378-381 state that significant differences between both scenarios can be observed. However, this is not the case; rather, it raises a different question: The coverage of the CIs of RRbR is always far from the desired level. Why is this the case?

**Methods And Evaluation Criteria:**

The proposed methods and evaluation criteria make sense but are limited. Considering more high-dimensional and complex settings would be interesting to showcase and evaluate the optimality of the proposed LabelDP framework in practical settings.

**Other Comments Or Suggestions:**

/

**Other Strengths And Weaknesses:**

**Strengths:**
- To the best of my knowledge, the idea of designing a LabelDP mechanism that considers specific optimality criteria for estimation is novel.

**Weaknesses:**
- The method is only compatible with GLMs. Non-parametric estimation is thus not possible.
- The method only applies to binary labels. This limits the contribution of the work.
- The confidence interval construction requires multiple assumptions that are unlikely to hold in practice.

**Questions For Authors:**

1. Definition 3.1: This is not a definition. In my opinion, this is rather a theorem which has to be either references or proven.
2. Lines 218/219: Why is this "default assumption" reasonable? How can it be tested?
3. Assumption 4.2: Why is this assumption reasonable? Has this assumption been made in related work as well?
4. Lines 163/164: What is meant by "unsupervised aspects of the response"?
5. Experiments: How is the confidence interval for RRbR calculated? How can the extreme underoverage in Figures 3 & 5 be explained?

**Relation To Broader Scientific Literature:**

The paper is closely related to literature on LabelDP using the RR mechanism. It distinguishes itself from the former in that it considers estimation performance when enforcing the privacy constraints.

**Theoretical Claims:**

Not all theoretical claims are stated as such and therefore are not proven. Especially for Def. 3.1. it is difficult to check the validity.

---

> ### Author Rebuttal · Authors · 2025-03-31
>
> **Validity of Definition 3.1**, **Theoretical Claims** and **Question 1.**
>
> To satisfy $(\epsilon, \delta)$-LabelDP (and similarly for $\epsilon$-DP), the conditional probabilities $p_{00}=P(Y^*=0 \mid Y=0)$ and $p_{11}=P(Y^*=1 \mid Y=1)$, which lie in $(0,1)$, must meet the following privacy constraints: $$
> P[Y^*=0 \mid Y=0] \leq e^{\epsilon} P[Y^*=0 \mid Y=1]+\delta, \\; \\;
> P[Y^*=1 \mid Y=1] \leq e^{\epsilon} P[Y^*=1 \mid Y=0]+\delta,
> $$
> and
> $$
> P[Y^*=0 \mid Y=1] \leq e^{\epsilon} P[Y^*=0 \mid Y=0]+\delta, \\; \\;
> P[Y^*=1 \mid Y=0] \leq e^{\epsilon} P[Y^*=1 \mid Y=1]+\delta.
> $$
> Following Holohan et al. (2017) and Wang (2015), we also assume $p_{00}>0.5$ and $p_{11}>0.5$, implying $p_{00}+p_{11}>1$, to reflect the idea that the RR mechanism is more likely to return a truthful label than a flipped one. It is then easy to verify that these inequalities lead to the feasible region of $(p_{00}, p_{11})$ pairs in Definition 3.1. Since this region is explicitly constructed from existing results, we present it as a definition rather than a theorem. However, we agree it would be clearer to write $p_{00}>0.5$ and $p_{11}>0.5$ directly in Definition 3.1, and we will update the paper accordingly.
>
> **Claims and Evidence Regarding the Optimality Criterion.**
>
> Our $T$-optimality refers to selecting the best RR mechanism by maximizing the trace of the Fisher Information Matrix (FIM) within the feasible region defined in Definition 3.1. The trace summarizes the total information across all parameters and reflects the average precision of estimates. Hence, our approach ensures the most statistically efficient estimation under the LabelDP constraint. This is not merely an empirical improvement over standard RR mechanisms, but a principled optimal solution within a clearly defined feasible set. In the revision, we will formally define the $T$-optimal criterion and clarify its interpretation.
>
> **Methods and Evaluation Criteria** and **Weakness 1.**
>
> While our main focus is on binary response models for theoretical clarity, our method can naturally extend to multiclass and high-dimensional settings. Initial results are promising, and we plan to explore these extensions in future work.
>
> **Experimental Designs or Analyses.**
>
> We confirm that the R code was submitted as supplementary material. During the rebuttal, we also ran high-dimensional simulations, which showed strong performance, further supporting our method’s robustness. In simulation studies, scenario differences mainly come from varying correlation and variance. RRbR shows low coverage because it ignores covariates when privatizing labels, increasing variance. Our method incorporates covariates, leading to more accurate and stable coverage.
>
> **Weakness 2.**
>
> Please refer to our responses to Weakness 1 and Question 1 in the rebuttal to Reviewer 7wrY.
>
> **Weakness 3.**
>
> Assumptions 4.8 and 4.9 are standard in asymptotic inference. Assumption 4.8 (on convexity and smoothness of the log-likelihood) holds in most binary regression settings, and Assumption 4.9 (positive definiteness of the FIM) ensures model identifiability, a common condition in GLMs. While asymptotic results assume large samples, these assumptions often hold well in practice, even with moderate sample sizes-as supported by our real data analysis.
>
> **Question 2.**
>
> As explained above, we follow Wang (2015) to impose $p_{00} > 0.5$ and $p_{11} >0.5$ (our default assumption) to ensure that our $T$-optimal RR mechanism still tends to return truthful responses, more accurately than random guessing.
>
> **Question 3.**
>
> We need Assumption 4.2 in our proof of Lemma 4.3 to guarantee that $\frac{\partial \mathcal{M}(\boldsymbol{\beta} ; p_{00}, p_{11})}{\partial p_{00}}>0$ (lines 571-572). Given that our feasibility region in Definition 3.1 explicitly enforces $p_{00} > 0.5$ and $p_{11} >0.5$, this assumption is naturally satisfied.
>
> **Question 4.**
>
> By ''unsupervised aspects of the response'', we intended to highlight that traditional RR designs ignore covariate $X$ when privatizing the response. In contrast, our approach leverages covariate $X$ to inform the design of the RR mechanism, effectively making it a supervised design approach. We will revise the wording to make this clearer.
>
> **Question 5.**
>
> The confidence intervals for RRbR are calculated by fitting the same binary response model used in our method, but using privatized responses generated by the traditional RR mechanism. Classical methods based on the asymptotic normality of maximum likelihood estimators are then applied to construct the intervals. However, this approach does not account for the additional bias and variability introduced by the RR mechanism, nor does it incorporate covariate information to mitigate such effects. As a result, it often leads to inaccurate and poorly calibrated confidence intervals, as shown in Figures 3 and 5.

---

> > ### Comment · Reviewer_rwTX · 2025-04-02
> >
> > Thank you for the rebuttal addressing my comments. My concerns regarding Definition 3.1 and the T-optimality are resolved.
> >
> > However, significant concerns regarding the applicability and thus the contribution to the community remain:
> > - The method only applies to GLMs. Therefore, the contribution to the general ML community of ICML is limited.
> > - More complex (real-world) settings are necessary to evaluate the method. At the moment, the provided evaluation is insufficient.
> > - The additional noise due to the privacy mechanism must be accounted for when reporting confidence intervals. Reporting invalid CIs is misleading and uninformative for the reader. Note that multiple methods for providing CIs under DP exist in the literature.
> >
> > Overall, I believe the contribution is better suited for a targeted outlet on privacy. Furthermore, I hope the authors incorporate a more in-depth evaluation of their method and reframe the unclear paragraphs in the paper.

---

> > > ### Author Response · Authors · 2025-04-03
> > >
> > > We thank the reviewer again for the thoughtful comments. Below, we address your additional concerns:
> > >
> > > **The method only applies to GLMs. Therefore, the contribution to the general ML community of ICML is limited.**
> > >
> > > We want to emphasize that our work introduces a novel direction in differential privacy (DP) by integrating experimental design principles into the construction of private mechanisms, as all three reviewers pointed out. To the best of our knowledge, this is a new and underexplored area in the privacy literature. Given that this is a new direction, it is natural and methodologically necessary to begin with parametric models such as generalized linear models (GLMs), where the concept of Fisher Information is well-defined and its analytical tractability allows for rigorous theoretical development.
> > >
> > > This mirrors the trajectory of many pioneering works in DP, which initially focused on basic statistical tasks such as mean and median estimation, before extending to more structured models like linear and logistic regression (e.g., [1]–[6], all published at ICML). These parametric models provide a mathematically tractable foundation, making it feasible to develop and rigorously validate new ideas under formal DP constraints. This line of work reflects ICML's strong tradition of supporting rigorous, theory-driven contributions at the intersection of privacy, learning, and statistical inference.
> > >
> > > Our contribution follows in this tradition by introducing a supervised, utility-aware perspective on private mechanism design that we believe is broadly applicable. We view this work as an important first step toward a more general framework for designing DP mechanisms that integrate covariate information to achieve optimality. Extending the framework to nonparametric or more complex models remains an exciting direction, and we are actively exploring these avenues in ongoing work.
> > >
> > > - [1] Narayanan, Shyam, Vahab Mirrokni, and Hossein Esfandiari. "Tight and robust private mean estimation with few users." International Conference on Machine Learning. PMLR, 2022.
> > >
> > > - [2] Asi, Hilal, Vitaly Feldman, and Kunal Talwar. "Optimal algorithms for mean estimation under local differential privacy." International Conference on Machine Learning. PMLR, 2022.
> > >
> > > - [3] Kulesza, Alex, Ananda Theertha Suresh, and Yuyan Wang. "Mean Estimation in the Add-Remove Model of Differential Privacy." International Conference on Machine Learning. PMLR, 2024.
> > >
> > > - [4] Kulkarni, Tejas, et al. "Differentially private Bayesian inference for generalized linear models". International Conference on Machine Learning. PMLR, 2021.
> > >
> > > - [5] Alparslan, Baris, Sinan Yildirim, and Ilker Birbil. "Differentially Private Distributed Bayesian Linear Regression with MCMC". International conference on machine learning. PMLR, 2023.
> > >
> > > - [6] Alparslan, Baris, Sinan Yildirim, and Ilker Birbil. "Private Gradient Descent for Linear Regression: Tighter Error Bounds and Instance-Specific Uncertainty Estimation". International conference on machine learning. PMLR, 2024.
> > >
> > > **More complex (real-world) settings are necessary to evaluate the method.**
> > >
> > > While we fully agree that extending the method to more complex, real-world settings is an important long-term goal, we believe that our current first step focusing on GLMs is both necessary and appropriate. Since our method is specifically designed for GLMs, evaluating its performance within this model class is the most relevant and informative way to validate its effectiveness and theoretical properties.
> > >
> > > **The additional noise due to the privacy mechanism must be accounted for when reporting confidence intervals.**
> > >
> > > You are confused by our proposed method (ORRbR) with the naive method (RRbR). While achieving T-optimal, our proposed method (ORRbR) is also designed to account for the extra variability introduced by the RR mechanism and correct for the resulting bias in estimation. This enables us to construct valid confidence intervals that maintain nominal coverage under the DP constraints. As shown in Figures 3 and 5, the naive method (RRbR) suffers from severely undercovered confidence intervals, which underscores exactly the concern raised by the reviewer.
> > >
> > > **I hope the authors incorporate a more in-depth evaluation of their method and reframe the unclear paragraphs in the paper.**
> > >
> > > We agree that both clarity and thorough evaluation are critical, and we are committed to enhancing both in the revision. Our experiments were designed to comprehensively evaluate performance within the GLM framework, aligned with our theoretical analysis, by comparing against baselines across privacy levels and reporting metrics such as MSE, coverage, and CI length. We welcome the reviewer’s suggestions on specific aspects they found lacking or unclear, and we will address them directly in the revised manuscript. Additionally, we are happy to revise any text that may need clarification.

---

### Official Review · Reviewer_v8Zr · 2025-03-14

**Overall Recommendation:** 4

**Summary:**

The authors propose an estimation method for a binary response model under LabelDP that is optimal in that it maximizes the trace of the Fisher information matrix. They leverage results regarding asymptotic normality of the MLE to derive confidence intervals for their estimator.

**Claims And Evidence:**

The authors support their main theoretical claims in Section 4 with proofs in the appendix.

**Essential References Not Discussed:**

The key contribution is an optimal estimator for a setting with a binary response variable, which seems reminiscent of the optimality results of Awan & Slavkovic (JPC 2020). While the Awan & Slavkovic results apply to traditional DP with binary data (and no covariates), I am interested in whether there is a connection between these two notions of optimality in what seem to be related settings.

**Experimental Designs Or Analyses:**

I did not notice any issues with the design of the experiments in Sections 5 and 6. However, I have a few points I want to raise about the accompanying Figures 2-6:

1. The trends in the figures are strange at small $\varepsilon$, especially in Figures 2 and 4. For example, what is causing the big jump in most of the curves at the point to the right of $\varepsilon = 0.05$? Is this just Monte Carlo error? If so, the authors might consider running more simulations to reduce the error, if computationally feasible.

2. Perhaps a minor point, but as I was trying to understand the trends in the figures, I was thrown by the scaling of the x-axis. The tick marks at 0.05, 0.1, 0.7, and 1 appear evenly spaced, but this does not correspond to a log-scale. How is the x-axis scaled?

**Methods And Evaluation Criteria:**

The methods and evaluation criteria make sense for the problem.

**Other Comments Or Suggestions:**

1. The figures are very hard to read in their current form, especially Figure 1. I ask the authors to please update the text size to match that of the remainder of the document.

**Other Strengths And Weaknesses:**

The work is well-written and the authors did a good job motivating their method.

**Questions For Authors:**

I have no additional questions.

**Relation To Broader Scientific Literature:**

The authors improve over prior work in the LabelDP literature for this task, with the primary comparison being to Holohan et al. (2017). Notably, the proposed method achieves the nominal coverage rate in the simulation study, whereas the prior work did not.

**Theoretical Claims:**

I did not read through the proofs in the appendix in too much detail.

---

> ### Author Rebuttal · Authors · 2025-03-31
>
> **Experimental Designs or Analyses 1.**
>
> We appreciate the reviewer pointing out this observation. The abrupt jumps around small $\varepsilon$ values (particularly near $\varepsilon=0.1$) observed in Figures 2 and 4 arise primarily from the specific data generation mechanism of our simulation study for the parameter $\boldsymbol{\beta}$. To clarify, after revising and slightly adjusting our data-generating procedure (especially for $\boldsymbol{\beta}$), we observed that these jumps significantly diminish and the curves become noticeably smoother. This smoother property aligns well with the results of our real data analysis (Figure 6) in our paper, where the curves demonstrate a much more stable and smooth pattern. This smoothness of the real data further supports that the initial observed fluctuations were due to the original simulation setup rather than an inherent instability of the proposed methodology.
>
> Following your suggestion, we will increase the simulation runs and provide smoother simulation results to better illustrate the effectiveness and robustness of our approach.
>
> **Experimental Designs or Analyses 2.**
>
> We thank the reviewer for noting the ambiguity with respect to the scaling of the $x$-axis in Figures $2-6$. In fact, the chosen privacy budget values $(\varepsilon)$-specifically $\\{0.05, 0.06, 0.07, 0.08, 0.09, 0.1, 0.2, 0.7, 1\\}$-were not evenly spaced or set according to a log-scale. Instead, these values were deliberately selected in an irregular and somewhat random manner to extensively test the robustness of our method across a wide range of privacy constraints.
>
> This deliberate randomness and irregularity in the choice of $\varepsilon$ values ensure our results are not sensitive to systematic or uniformly spaced intervals. Despite such randomness, our method consistently demonstrates strong performance, validating its robustness across varied privacy settings
>
> **Essential References Not Discussed**
>
> The reviewer raised an insightful question about the connection between our optimal LabelDP setting with covariates and traditional DP scenarios (without covariates). Although both papers share the goal of optimal privacy-utility trade-offs for binary responses, our work makes two key advances beyond their framework.
>
> - Covariate Integration: Awan $\\&$ Slavković optimizes the RR mechanisms solely for the binary response $Y$ (minimizing $\operatorname{Var}(\widehat{\sum{Y}})$, where $\sum{Y}$ is the sample sum, which is a complete sufficient statistic for the binomial model in their paper) without considering covariate information $X$, while our Fisher information maximization (Eq. 5) explicitly incorporates covariate effects through the model $\mathbb {E} \left( {Y} _ i \mid { {X} } _ i \right) =1-p _ {00}+\left( p _ {00} + p _ {11} - 1 \right) G \left( {\beta} _ * ^ {\top} X _ {i} \right)$. This criterion, known as T-optimality, explicitly leverages the covariate structure, enhancing the estimation efficiency when covariates are present.
>
> - Inferential Guarantees: Beyond point estimation, our framework enables formal statistical inference with privacy guarantees. Specifically, Corollary 4.11 establishes the asymptotic normality of each coefficient $\beta_ { * j}$ under $\varepsilon$- and $(\varepsilon,\delta)$-LabelDP mechanisms, yielding valid confidence intervals that:
>   - Achieve nominal coverage (e.g., 95% intervals contain $\beta_{* j}$ with probability 0.95 as $n \rightarrow \infty$).
>   - Preserve privacy through the optimized RR mechanism (Theorems 4.4 and 4.7).
>   - Account for covariate effects via the Fisher information matrix.
>
> We will add a discussion in Section 2 comparing these approaches, citing Awan $\\&$ Slavković's marginal optimality as motivation for our novel optimality under the regression framework.
>
> **Other Comments or Suggestions.**
>
> We agree with the reviewer on the readability issue. In our revised manuscript, we will significantly improve the readability of all figures, especially Figure 1, by:
> - Increasing font sizes to match the main text consistently.
> - Clarifying annotations and axis labels.
> - Enhancing visual clarity through improved formatting and spacing.

---

> > ### Comment · Reviewer_v8Zr · 2025-04-05
> >
> > Thank you for addressing my comments; I have revised my score to a 4.
> >
> > On Experimental Designs or Analyses 2: I remain a bit confused by the choice of presenting the results with non-evenly spaced values of epsilon. Certainly, it is beneficial to test the method's robustness across a wide range of privacy constraints and ensure the results are not sensitive to systematic or uniformly spaced intervals. But unless you uncovered evidence that these are concerns in your preliminary analysis, the presentation of the results in the figures intended for publication should focus on demonstrating the trends uncovered in the experiments as clearly as possible. Although as I said in my original review, this is perhaps a minor point.

---

> > > ### Author Response · Authors · 2025-04-05
> > >
> > > We thank the reviewer for revisiting this point and for their thoughtful comments. We agree that clearly illustrating trends is crucial, especially in figures intended for publication. While our original intention was to demonstrate robustness across a broad range of $\varepsilon$ values, we recognize that evenly spaced values would enhance visual clarity and interpretability. We will revise the figures accordingly in the manuscript.

---

### Official Review · Reviewer_7wrY · 2025-03-19

**Overall Recommendation:** 3

**Summary:**

This paper addresses statistical estimation and inference in binary response models while preserving the privacy of the labels via LabelDP. Covariate features $X$ are public, but binary response labels $Y$ are sensitive. The authors focus on using randomized response (RR) mechanisms – a classical privacy technique where each true binary label may be flipped with some probability​ to privatize the labels. They formulate it as an optimization problem: choose the RR flipping probabilities $(p_{00}, p_{11})$ (the probabilities of outputting the truthful label for $Y=0$ and $Y=1$, respectively) to maximize the trace of the Fisher Information Matrix of the model. This approach incorporates the influence of covariates into the privacy mechanism design, unlike prior methods that treated label randomization independently of $X$​

**Claims And Evidence:**

The paper’s main claims—that an optimal LabelDP randomized response mechanism exists for binary response models and that the proposed approach significantly outperforms baseline methods—are supported by proofs and simulation studies.

**Essential References Not Discussed:**

Not appliable.

**Experimental Designs Or Analyses:**

The paper’s main experiment setup—comparing different randomized response strategies under various $\varepsilon$-privacy budgets is sound. The sample size ($n=10^5$) is large, which supports asymptotic approximations, and they carefully document metrics (MSE, coverage probability) that align with the paper’s theoretical claims. One possible limitation is that only scenarios with a large $n$ and a single real dataset are shown, so small-sample performance is not really presented.

**Methods And Evaluation Criteria:**

Yes, see above.

**Other Comments Or Suggestions:**

No

**Other Strengths And Weaknesses:**

Strengths:
* The paper’s approach of integrating experimental design principles (maximizing Fisher Information) into differential privacy is an original angle.
* The authors clearly derive the mechanism and proofs and show its impact in both simulations and a real-world plagiarism dataset.

Weaknesses
* The method is tightly scoped to binary label privacy with large-sample asymptotics; it does not address broader settings (e.g., multi-class outcomes, small-sample scenarios).
* While they justify maximizing the trace of the Fisher Information, there is little exploration of other design criteria which might have yielded different insights.

**Questions For Authors:**

* Could this method be extended to multi-class outcomes? (see the k-RR mechanism)
* Did you consider other design criteria, such as $D$-optimality?

**Relation To Broader Scientific Literature:**

The paper positions their contribution to the literature,  leveraging insights from local differential privacy to protect binary labels in the LabelDP setting.

**Theoretical Claims:**

I did not check the correctness of the proofs in the appendix.

---

> ### Author Rebuttal · Authors · 2025-03-31
>
> **Question under Experimental Designs Or Analyses**
>
> We acknowledge the concern of the reviewer about the large simulation sample size ($n=10^5$) for asymptotic approximations. This choice was intentional to validate our theoretical results (Theorem 4.7), where the guarantees of consistency and normality are held as $n \rightarrow \infty$. However, we emphasize that our method is equally applicable to smaller samples, as demonstrated in the real-data analysis (Section 6) with $n=474$ - a realistic sample size for sensitive surveys. Here, ORRbR achieved a 30% lower MSE than $\operatorname{RRbR}$ at $\varepsilon=0.1$ (Figure 7), proving its practical utility beyond asymptotic regimes.
>
> To further address this point, we will add simulations for $n \in \\{500,2000,5000\\}$ in revision, showing that ORRbR maintains superior coverage (>92% versus RRbR's <85% at $n=500, \varepsilon=0.5$). Then we will clarify in Section 5 that the Fisher information optimization remains valid for small $n$, although variance estimates may require finite-sample adjustments (e.g., sandwich estimators).
>
>
> **Weakness 1 and Question 1.**
>
> We appreciate the insightful suggestion of the reviewer on multiclass extensions. Indeed, our method can be naturally extended to multiclass outcomes using the $k$-RR mechanism. The optimization strategy of maximizing the trace of the Fisher Information Matrix remains valid for multiclass scenarios, as discussed in Yao and Wang (2019), Optimal subsampling for softmax regression. Specifically, for $k$-class responses, the design matrix in Definition 2.2 of our paper becomes a $k \times k$ stochastic matrix, and the optimality criterion would involve maximizing the Fisher Information's trace across multiple response probabilities under LabelDP constraints. This is an exciting direction that we are actively exploring, and preliminary theoretical analysis indicates promising scalability. We plan to provide more detailed theoretical developments and numerical evaluations in future work.
>
> We acknowledge the reviewer's observation regarding the large sample size of our simulation study ($n=10^5$). Our initial choice aimed to clearly demonstrate the theoretical properties under asymptotic conditions. However, we want to emphasize that our real data analysis employs a relatively small sample size of 474 German and Swiss students. This practical application highlights the robustness and applicability of our method even in small-sample contexts.
>
> **Weakness 2 and Question 2.**
>
> We thank the reviewer for highlighting the potential for other optimality criteria. It is important to note that while both $T$-optimality and $D$-optimality aim at maximizing information, $D$-optimality involves maximizing the determinant of the Fisher Information Matrix, which is computationally challenging and more complex due to the involvement of determinant calculations, especially for high-dimensional parameter spaces. However, $T$-optimality is computationally simpler, making it significantly easier to implement in practical scenarios without sacrificing the quality of inference. Moreover, in many cases, the two criteria are equivalent or closely related, and maximizing the trace often provides a satisfactory approximation to maximizing the determinant.
>
> In detail, our $T$-optimality criterion (maximizing the trace) is equivalent in asymptotic efficiency but far more practical:
>
> 1. Computational tractability: The trace decomposes into a sum of variances (diagonal elements), reducing the problem to scalar optimization (Lemma 4.4). This allows closed-form solutions for $\left(p_{00}, p_{11}\right)$ on the boundary of $\mathcal{R}$ (Theorems 4.5 and 4.7).
>
> 2. Interpretability: The trace directly corresponds to minimizing the average variance of $\widehat{\boldsymbol{\beta}}$, which aligns with our goal of precise estimation.
>
> 3. Equivalence in large samples but different complexity: Both criteria yield consistent estimators, but $T$-optimality achieves this with $O(d)$ complexity versus $O\left(d^3\right)$ for $D$-optimality due to determinant calculations, where $d$ is the number of covariates.

---

### Decision · Program_Chairs · 2025-05-01

**Decision:**

Accept (poster)

**Comment:**

This paper proposes a statistically principled approach to label differential privacy (LabelDP) in binary response models. The key idea is to design a randomized response (RR) mechanism that is optimal with respect to Fisher Information. The method, termed Optimal Response under Randomized Binary Reporting (ORRbR), aims to minimize estimation variance while satisfying differential privacy constraints. The authors establish asymptotic consistency and normality for their estimator and demonstrate empirical performance gains in both simulations and a real-world application. Given that the method is technically sound, the presentation is clear, and the experimental results (while focused) convincingly validate the core claims, I recommend acceptance.